# Injectables' key role in rifampicin-resistant tuberculosis shorter treatment regimen outcomes

Tom Decroo[1,2]*, Aung Kya Jai Maug[3], Mohamed Anwar Hossain[4], Cécile Uwizeye[1], Mourad Gumusboga[1], Tine Demeulenaere[5], Nimer Ortuño-Gutiérrez[5], Bouke C. de Jong[1], Armand Van Deun[6]

1 Institute of Tropical Medicine, Antwerp, Belgium, 2 Research Foundation Flanders, Brussels, Belgium, 3 Damien Foundation Bangladesh, Dhaka, Bangladesh, 4 Family Health International, Rajshahi, Bangladesh, 5 Damien Foundation Brussels, Brussels, Belgium, 6 Independent Consultant, Leuven, Belgium

* tdecroo@itg.be

**Data Availability Statement:** The data supporting the findings of this publication are retained at the Institute of Tropical Medicine, Antwerp and will not be made openly accessible due to ethical and

## Abstract

### Background

Since a meta-analysis showed little or no effect of second-line injectables on treatment success, and using injectables may induce ototoxicity, injectable-free rifampicin-resistant tuberculosis (RR-TB) treatment regimens are recommended. However, acquired resistance preventing activity was overlooked. No previous study assessed the effect of shortening the duration of kanamycin administration to 2 months during the intensive phase of the RR-TB shorter treatment regimen (STR).

### Methods

Retrospective cohort study of the effect of using 2 months of kanamycin instead of the standard 4(+) months (extension if lack of smear conversion at 4 months) on recurrence (either treatment failure or relapse) and fluoroquinolone acquired drug resistance, in patients treated with a gatifloxacin-based STR in Damien Foundation supported clinics in Bangladesh. Logistic regression was used to estimate associations.

### Results

Five of 52 (9.6%) treated with a STR containing two months of kanamycin had recurrence, compared to 21 of 738 (2.8%) patients treated with the standard STR containing 4(+) months of kanamycin (OR 3.7; 95%CI:1.5–10.3). In those with initially fluoroquinolone-susceptible TB, acquired resistance to fluoroquinolone was detected in none of 639 patients treated with 4(+) months of kanamycin and two (4.5%) of 44 treated with two months of kanamycin (OR 75.2; 95%CI:3.6–1592.1).

### Conclusion

Two months of kanamycin was insufficient to prevent recurrence with acquired resistance to gatifloxacin, the core drug of the most effective RR-TB STR. Injectable mediated resistance

privacy concerns. Data can however be made
available after approval of a motivated and written
request to the Institute of Tropical Medicine at
ITMresearchdataaccess@itg.be.

**Funding:** The authors received no specific funding
for this work.

**Competing interests:** The authors have declared
that no competing interests exist.

prevention is important to reach high effectiveness, to safeguard all treatment options after
recurrence, and to prevent the spread of resistant TB. Studies on all-oral regimens should
also assess the effect of regimen composition on resistance acquisition. Until evidence
shows that other drugs can assure at least the same strong resistance preventing activity of
the injectables, it seems wise to continue using this group of drugs, and adapt the regimen if
any ototoxicity is detected.

## Introduction

Since the end of 2019, the World Health Organization (WHO) recommends to use an
injectable-free shorter treatment regimen for the treatment of rifampicin-resistant tuberculosis
(RR-TB) [1]. However, this recommendation has a "very low certainty in the estimate of the
effect" from a 2018 meta-analysis, and is conditional, "recognizing that different choices will
be appropriate for individual patients" [2]. In fact, results of ongoing comparative trials, such
as STREAM II where bedaquiline is substituted for kanamycin in one of the arms, have not yet
been published. The recommendation was informed by unpublished results from a South African
cohort study and the 2018 metanalysis comparing the effect of drugs but not of regimens,
ignoring outcomes such as recurrence due to acquisition of resistance to the regimen's core
drug [3].

Since decades, second-line injectables, such as kanamycin, amikacin, and capreomycin
have been used for the treatment of RR-TB. A 2012 meta-analysis showed that treatment success
was most likely when second-line injectables were used during the first 8 months of treatment
in long treatment regimens [4]. A subsequent meta-analysis confirmed that initial
susceptibility to fluoroquinolone and the second-line injectables were the strongest predictors
of treatment success [5].

Between 1997–2007 the shorter treatment regimen (STR) was developed in Bangladesh.
Stepwise, six different short regimens were evaluated [6]. Modifications that may seem minor
had important consequences on the bacteriological effectiveness of the regimens, and showed
that constructing TB treatment regimens is more complex than combining "at least four likely
active drugs" [6, 7]. The most effective STR used a 4-month intensive phase. To avoid relapse
due to a high bacillary load this intensive phase was extended by a maximum of two months in
case smear-conversion was not achieved at four months. During the intensive phase kanamycin,
prothionamide, and high-dose isoniazid were added to a background regimen that
included high-dose gatifloxacin as the later generation fluoroquinolone, ethambutol, clofazimine
and pyrazinamide [6]. The STR also showed to be highly effective in Niger and Cameroon,
which also used gatifloxacin [8, 9]. Use of the STR was recommended in WHO's 2016
guidelines [7], and the first STREAM trial showed that its moxifloxacin-based variant was
non-inferior to 20-month long RR-TB regimens. The twenty-month regimens included an
intensive phase containing a second-line injectable for a minimum of 8 months [10].

Although the duration of second-line injectable administration was shortened from at least
eight months in long regimens to at least four months in the STR, the incidence of severe
adverse events, in particular hearing loss, warranted studies on regimens that further reduced
use of second-line injectables. For streptomycin, clearly suboptimal effectiveness had been
documented when it was used for only the first month in the 8-month first-line regimen, with
rifampicin and pyrazinamide for the first two months and isoniazid throughout (complemented
by thioacetazone after the first two months). Whether its use beyond the second

month improved the results was not assessed, possibly because there were zero recurrences after the regimen with streptomycin for two months, versus 7% relapse, half with acquired isoniazid resistance, if given during the first month only [11]. To our best knowledge trials on the optimal duration of second-line injectables for RR-TB have not been conducted. In this study we therefore document the assessed effect on bacteriological outcomes of shortening the duration of kanamycin administration from four to two months, without changing the duration for the other drugs used during the intensive phase. Bacteriological outcomes included recurrence (either failure or relapse) and resistance acquisition to gatifloxacin.

## Methods

### Study population

This retrospective study uses data collected during the evaluation of the STR in Bangladesh. Patients diagnosed with bacteriologically confirmed RR-TB and who started treatment with a gatifloxacin-based STR between December 2005 and October 2015 were included. The modified regimen, with 2 months of kanamycin during the 4(+)-month intensive phase was used in the second half of 2011 only. All other procedures, including monitoring, were the same. The composition of regimens is shown in Table 1. To assess the bacteriological effect of the STR, patients lost to follow-up and those who died were excluded.

### Data collection and analysis

Patient follow-up, data collection, and drug susceptibility testing methods were described in detail in a previously published paper [12]. Recurrence included treatment failure and relapse, diagnosed using active follow-up including 6-monthly cultures for two years after cure. Treatment outcomes are also shown in a previously published paper [13]. In short, treatment failure was defined as having a positive culture between the end of the 5th treatment month and the end of treatment. Relapse was defined as having at least one positive culture after the end of successful treatment. Results showing resistance, detected by any genotypic or phenotypic method, overrode results from another sputum showing susceptibility. Initial fluoroquinolone resistance was categorized as 'high-level' if the strain grew on agar or Löwenstein–Jensen medium at $\geq 8$ mg/L OFX or if the minimum inhibitory concentration (MIC) for GFX was $\geq 2$ mg/L. Resistance was defined as acquired when resistance was detected on the recurrence isolate but not present at baseline, and with DNA fingerprinting showing the same strain.

Bivariable logistic regression was used to assess the correlation between regimen (regimen with 4(+) months kanamycin versus 2 months kanamycin in the intensive phase), initial resistance to fluoroquinolone, and initial resistance to kanamycin and having a bacteriologically adverse outcome. Bacteriologically adverse outcomes included recurrence and acquired resistance to fluoroquinolone. The Kruskal–Wallis test was used to compare the median duration of the intensive phase between regimens.

**Table 1. Composition of gatifloxacin-based shorter treatment regimens used in Bangladesh.**

|  | Enrolment period | Intensive phase # | Continuation phase |
|---|---|---|---|
| Standard STR | 2005–2015 | 4(+) KCG$^h$EH$^h$ZP | 5 G$^h$EZC |
| Modified STR | 2011 sem. II | 2KCG$^h$EH$^h$ZP / 2(+) CG$^h$EH$^h$ZP | 5 G$^h$EZC |

C: clofazimine; E: ethambutol; G$^h$: high-dose gatifloxacin; H$^h$: high-dose isoniazid; K: kanamycin; P: prothionamide; Z: pyrazinamide

STR: shorter treatment regimen; sem. II: second semester

# The standard duration of the intensive phase was 4 months, but was extended until negative on sputum smear microscopy.[12]

## Ethics

The evaluation of the STR is fully covered by a study protocol approved by the National Research Ethics Committee of Bangladesh Medical Research Council and the Institutional Review Board of the Institute of Tropical Medicine Antwerp. All patients completed and signed an informed consent form in the local language before starting treatment.

## Results

Fig 1 shows enrolment in the study of patients treated with the standard gatifloxacin-based STR, with 4(+) months of kanamycin, or the modified STR, with 2 months of kanamycin. Of 928 patients treated with a gatifloxacin-based regimen, 790 were included in the analyses after exclusion of those who died or were lost to follow-up.

The distribution of sex and age was not significantly different among patients treated with the modified STR (2 months of kanamycin) compared to those treated with the standard STR (4(+) months of kanamycin) (Table 2).

Of 738 patients on the standard STR (4(+) months of kanamycin) and 52 on the modified STR (2 months of kanamycin), 717 (97.2%) and 47 (90.4%) were treated successfully, respectively. The median duration of the intensive phase was similar for both cohorts: 120 days (IQR 119–127) for those treated with the standard STR and 120 days (IQR 119–121) for those treated with the modified STR (p = 0.1).

Of 790 patients on a gatifloxacin-based STR, 52 were treated with a modified regimen, with two months of kanamycin (Table 3). Recurrence was detected in 21 of 738 (2.8%) patients treated with 4(+) months of kanamycin and five of 52 (9.6%) treated with two months of kanamycin. Using two months instead of 4(+) months of kanamycin was significantly associated with recurrence (OR 3.7; 95%CI: 1.5–10.3), as were low-level (OR 12.8; 95%CI: 3.1–53.4) and high-level (OR 107.2; 95%CI: 35.9–320.1) initial resistance to fluoroquinolone, and initial resistance to kanamycin (OR 31.7; 95%CI: 6.8–147.4).

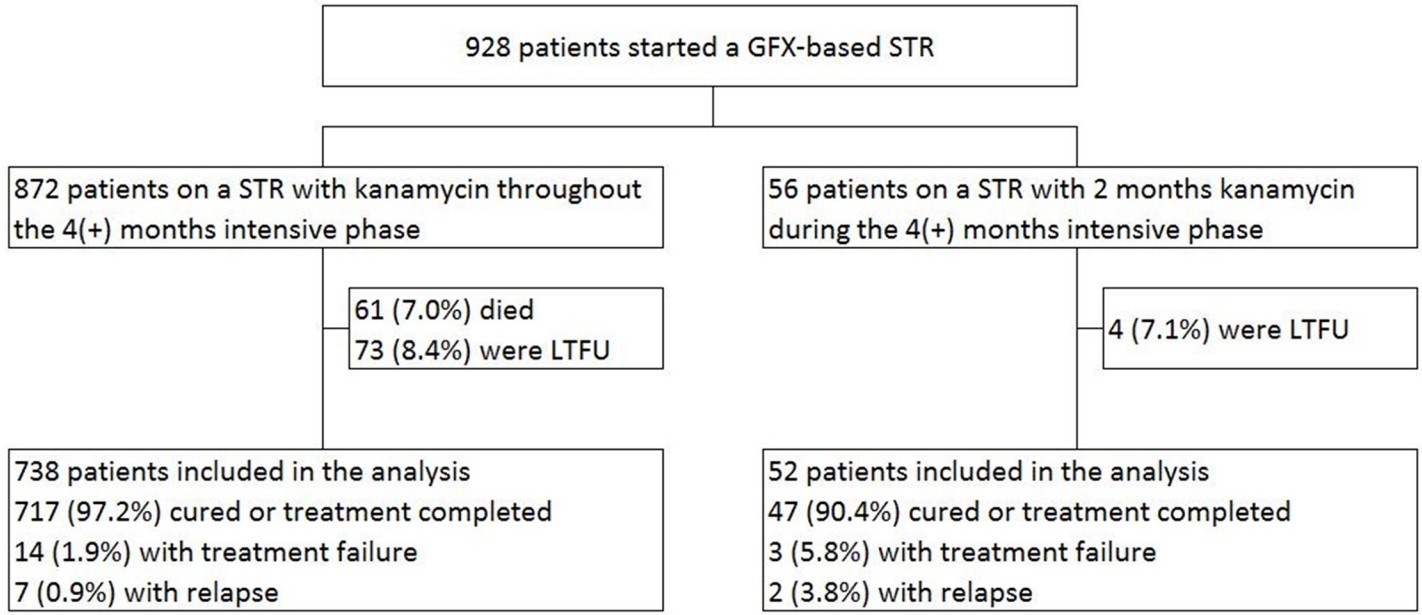

**Fig 1. Study enrolment and treatment outcomes among 928 patients treated with a gatifloxacin-based regimen in Bangladesh.** LTFU: lost to follow-up; GFX: gatifloxacin; STR: shorter treatment regimen.

**Table 2. Age and gender stratified by regimen, either 2 months or 4(+) months of kanamycin in 790 RR-TB patients treated with a gatifloxacin-based shorter treatment regimen in Bangladesh.**

| | Total | | Kanamycin during the intensive phase | | | | |
| --- | --- | --- | --- | --- | --- | --- | --- |
| | | | 4(+) months | | 2 months | | |
| | N | % | N | % | N | % | p-value |
| Total | 790 | | 738 | | 52 | | |
| **Sex** | | | | | | | 0.1 |
| Female | 243 | 30.8 | 222 | 30.1 | 21 | 40.4 | |
| Male | 547 | 69.2 | 516 | 69.9 | 31 | 59.6 | |
| **Age group** | | | | | | | 0.1 |
| < = 18 years | 76 | 9.6 | 68 | 9.2 | 8 | 15.4 | |
| >18–35 years | 409 | 51.8 | 378 | 51.2 | 31 | 59.6 | |
| >35–50 years | 207 | 26.2 | 199 | 27 | 8 | 15.4 | |
| >50 years | 98 | 12.4 | 93 | 12.6 | 5 | 9.6 | |

In 189 of 738 patients on the standard STR, the intensive phase (with additional high-dose isoniazid, ethionamide and kanamycin) was extended. Of 189, 10 (5.3%) experienced recurrence (8 treatment failure, 2 relapse, but without acquisition of resistance to fluoroquinolone). In two of 52 patients on the modified regimen, the intensive phase (with additional high-dose isoniazid and ethionamide, but not with kanamycin) was extended beyond 4 months, but without success (both experienced treatment failure with acquisition of resistance to fluoroquinolone, and later died of active TB because an adequate regimen could not be constituted or was refused by the patient).

**Table 3. Effect of using 2 months of kanamycin instead of 4(+) months on recurrence in 790 RR-TB patients treated with a gatifloxacin-based shorter treatment regimen in Bangladesh.**

| | Total | Recurrence $ | | OR # | [95%CI] |
| --- | --- | --- | --- | --- | --- |
| | N | N | % | | |
| **Total** | **790** | **26** | **3.3** | **NA** | |
| **Kanamycin during the intensive phase** | | | | | |
| 4(+) months | 738 | 21 | 2.8 | 1 | |
| 2 months | 52 | 5 | 9.6 | 3.7** | [1.5,10.3] |
| **Initial fluoroquinolone resistance** | | | | | |
| Susceptible | 684 | 4 | 0.6 | 1 | |
| Resistant low-level | 44 | 3 | 6.8 | 12.8*** | [3.1,53.4] |
| Resistant high-level | 46 | 19 | 41.3 | 107.2*** | [35.9,320.1] |
| Missing | 16 | 0 | 0 | NA | |
| **Initial kanamycin resistance** | | | | | |
| Susceptible | 767 | 23 | 3 | 1 | |
| Resistant | 6 | 3 | 50 | 31.7*** | [6.8,147.4] |
| Missing | 17 | 0 | 0 | NA | |

* p<0.05

** p< 0.01

*** p<0.001

$ Recurrence: either treatment failure or relapse

# Firth logistic regression was used as events were rare. For the same reason no multivariable model was developed.

**Table 4. Effect of using 2 months of kanamycin instead of 4(+) months on recurrence and fluoroquinolone acquired drug resistance in 683 patients with initially fluoroquinolone-susceptible/kanamycin-susceptible RR-TB, treated with a gatifloxacin-based shorter treatment regimen in Bangladesh.**

| | Total | | Recurrence $ | | | | Fluoroquinolone acquired drug resistance £ | | |
|---|---|---|---|---|---|---|---|---|---|
| | | N | % | OR # | [95%CI] | N | % | OR # | [95%CI] |
| Kanamycin during the intensive phase | | | | | | | | | |
| 4(+) months | 639 | 2 | 0.3 | 1 | | 0 | 0 | 1 | |
| 2 months | 44 | 2 | 4.5 | 15.0** | [2.5,89.0] | 2 | 4.5 | 75.2** | [3.6,1592.1] |

** p< 0.01

$ Recurrence: either treatment failure or relapse

£ Fluoroquinolone acquired drug resistance: fluoroquinolone resistance in recurrence sample for patients with initially fluoroquinolone-susceptible tuberculosis in the same strain as shown by DNA fingerprinting

# Firth logistic regression was used as events were rare. For the same reason no multivariable model was developed.

None of 52 patients treated with the STR containing two months of kanamycin had clinical signs of hearing loss (audiometry was not in place). Data on clinical monitoring was available for 515 patients treated with the standard STR and showed hearing loss in 26 (5.0%) patients.

Of 683 patient with initially fluoroquinolone-susceptible and kanamycin-susceptible TB, 44 were treated with a modified regimen, with two months of kanamycin (Table 4). Acquired resistance to fluoroquinolone was detected in none of 639 patients treated with 4(+) months of kanamycin and two (4.5%) of 44 treated with two months of kanamycin. Using two months instead of 4(+) months of kanamycin was associated with recurrence (OR 15.0; 95%CI: 2.5–89.0) and fluoroquinolone acquired drug resistance (OR 75.2; 95%CI: 3.6–1592.1).

## Discussion

In this study we compared two STR regimens: a standard STR including kanamycin throughout the 4(+) month intensive phase, and a modified STR that included 2 months of kanamycin during the 4(+) month intensive phase. The only difference between both regimens was the duration of kanamycin administration. The aim to avoid ototoxicity by halving the amount of injectable may have been reached. Clinical monitoring did not reveal a single case among the 52 patients of the 2 month kanamycin cohort, compared to the 5% in the cohort using kanamycin throughout the intensive phase, also monitored clinically, or the 7% of serious hearing loss reported for the African STR study that used audiometry [14]. However, the frequency of bacteriologically adverse outcomes, such as recurrence and acquired resistance to the core drug, were significantly more frequent when kanamycin administration was limited to two months. These results are coherent with those from a trial in patients with rifampicin-susceptible TB, which also showed a higher frequency of recurrence and acquired resistance to other drugs in the regimen when the duration of the injectable (streptomycin in this trial) was reduced from 2 to one month [11]. Shortening the duration of injectables in RR-TB treatment regimens was not yet studied.

The gatifloxacin-based STR is the first treatment regimen that consistently achieved high RR-TB treatment success as well as very low recurrence rates without creating any additional resistance to this core drug in a wide range of countries, and does so for more than a decade [15]. In contrast with the WHO recommended approach of combining "at least 4 likely active drugs" [2], the effectiveness of the STR stems from combining drugs with complementary activity [16]. Based on data from numerous clinical trials forming the cornerstone of current first-line regimens, the specific activities of different TB drugs accounting for the effectiveness

of a regimen were first described by the late Prof. Mitchison [17]. Maybe his most important conclusion was that, with susceptible bacilli, the efficacy of a combination of isoniazid, rifampicin and pyrazinamide could be enhanced only very little by the addition of streptomycin, and not at all by ethambutol [18]. Streptomycin also prevented acquired resistance when added to isoniazid, rifampicin and pyrazinamide in high risk patients with very high recurrence rates [19]. Both streptomycin and ethambutol were classified as companion drugs, indicated for prevention of acquired resistance to the more effective drugs. Streptomycin was rated to be stronger [18].

A multi-country study clearly showed that all three second-line injectables protect against acquired resistance to the driving fluoroquinolone core drug [20], and also the present study underpins that kanamycin plays this crucial role. In the STR a later generation fluoroquinolone (preferably gatifloxacin) [13] is the core drug, contributing most to achieving relapse-free cure through high bactericidal and high sterilizing activity. As only few drugs have this characteristic, core drugs need to be protected, to prevent resistance acquisition and the loss of a highly effective standardized regimen [16]. Isoniazid protects rifampicin in first-line treatment rifampicin-throughout regimens [21], whereas kanamycin protects the fluoroquinolone in the STR. Indeed, in patients without sputum smear conversion at 4 months, the intensive phase was prolonged with ethionamide, high-dose isoniazid and with kanamycin in addition to the background regimen when the standard STR was used, but without kanamycin in the modified STR (kanamycin was already stopped at month 2). Extending the intensive phase was not successful when kanamycin was not included. Both patients on the modified STR in whom the intensive phase was extended experienced treatment failure with acquisition or resistance to fluoroquinolone. Even though numbers were small for the modified STR cohort, this finding suggests that, given sufficient time, kanamycin succeeds in reducing numbers of mutants resistant to the core drug to a clinically irrelevant level, where other companion drugs with little bactericidal activity fail [16].

The 2018 meta-analysis declared that the second-line injectables kanamycin and capreomycin were not effective and that amikacin had a modest beneficial effect on treatment success. However, it did not assess acquired resistance preventing action [3]. Indeed, as for streptomycin in the original first-line regimens clinical trials, initial susceptibility to second-line injectables is not expected to significantly improve success. Also in our two-month kanamycin cohort, relapse-free cure was not lower than published for the 4(+) month kanamycin cohort [12] due to overall very low rates of adverse outcomes. The lack of second-line injectable early bactericidal activity (EBA) is often taken as an argument for their lack of activity. However, to the best of our knowledge only amikacin EBA values over not more than two days have been published. No difference with untreated controls could be shown, but the authors added that the bacterial fall seen in the controls was unprecedented [22]. Of interest, while the same group reported 2-day EBA close to zero for streptomycin standard dose (15 mg/kg), similar to the value reported for amikacin, EBA was significantly higher at double dose [23]. Besides EBA estimates being prone to considerable variation, assessment at two days treatment is not appropriate for all TB drugs [24]. Highly active TB drugs such as pyrazinamide, clofazimine and bedaquiline show their full EBA after a delay of about two weeks [25]. In the case of clofazimine this had led to contradicting findings, depending on the test system used [26]. As a result, it was recognized as a valuable TB drug only after its sterilizing power in mice had been demonstrated [27], although its activity against TB bacilli had been known for 60 years [28]. In mice, streptomycin and amikacin were highly bactericidal in the first months, but particularly streptomycin failed to complete sterilization [29]. In vitro the bacillary kill after one week was larger for streptomycin than for isoniazid or rifampicin [30]. Sophisticated in-vitro experiments showed a 2-log kill by amikacin in the first two days, and this for dormant as well as actively replicating TB bacilli [31].

In summary, these reports testify of an immediate very large and rapid kill of TB bacilli, irrespective of metabolic state, continuing though the early phase of treatment when numbers of still replicating resistant mutants are highest. This is highly consistent with the resistance preventing role assigned to streptomycin and the second-line injectables. Only amikacin was attributed an additional modest contribution to sterilization, which may explain why it was the only second-line injectable recognized as having an effect on treatment success by the 2018 meta-analysis [3]. Furthermore, effectiveness of these drugs strongly depends on peak serum concentration, the reason why pharmacologists recommend testing a higher dose, to achieve high serum peaks for optimal effectiveness, with intermittent instead of daily administration so as to reduce the total amount administered and thus toxicity [32]. As toxicity is correlated with the cumulative drug exposure to any second-line injectable during different treatment episodes [33], toxicity will be reduced further because streptomycin is no longer used in first-line retreatment regimens [34]. One could then imagine the return of a valuable drug that was discarded due to excess toxicity, simply by total dose reduction. In the past, pyrazinamide was abandoned because of excessively frequent hepatotoxicity [35], but it became one of the cornerstones of short-course chemotherapy, once it was found to be effective at close to half the original dose [36].

We believe that the second-line injectables have been discarded prematurely [1], based on rates of ototoxicity when they had to be used for at least eight months in patients who often had prior exposure to streptomycin [37], and with remarkable disregard of their essential contribution to a regimen in the context of the current new-TB-drug-euphoria. We strongly recommend that studies assessing the effect of injectable-free STR also assess the effect of this modification on bacteriological outcomes such as recurrence and resistance acquisition. Meanwhile, in patients with a contra-indication for the use of second-line injectables, such as those with baseline hearing impairment, those who develop adverse events, or those with initial resistance to second-line injectables, the second-line injectable can be replace by linezolid. Experience from Niger shows that this approach can prevent severe ototoxicity and that linezolid related adverse events were manageable. Similar cure rates were achieved in patients treated with the standard STR and in those treated with a modified STR, using linezolid instead of kanamycin [38].

Our study has several strengths and limitations. We were able to assess the effect of two months versus 4(+) months of kanamycin in the intensive phase on bacteriological outcomes as the regimens included in this study were the same, except for the duration of kanamycin administration. Using a continuously updated database all patients registered as having started treatment were included in the analysis, including that of relapse after cure. Our study also has some limitations. Patients were not randomly assigned to regimens. Moreover, the sample of patients treated with the modified regimen was small, because enrolment was stopped as soon as the second case of acquired fluoroquinolone resistance was reported, which had never been encountered with the unmodified gatifloxacin regimen. Both cases ultimately died of active TB because an adequate regimen could not be constituted or was refused by the patient. Excess mortality is too high a price to pay for preventing ototoxicity.

In conclusion, 2 months of kanamycin during the intensive phase of the RR-TB STR was insufficient to prevent recurrence with acquired resistance to high-dose gatifloxacin, the most powerful fluoroquinolone and core drug of the regimen. Our findings confirm the important role of second-line injectable drugs with regards to bacteriological outcomes. Future studies on all-oral regimens should not only assess composite endpoints, combining mortality, loss to follow-up, and recurrence, but also the effect of regimen composition on resistance acquisition. Moreover, these should also report on the composition and effectiveness of salvage regimens for patients who fail an all-oral regimen. Meanwhile, it may be wiser to continue

using a second-line injectable, replacing it by linezolid in case hearing impairment is identified.

## Acknowledgments

We thank the study participants and staff of Damien Foundation Bangladesh.

## Author Contributions

**Conceptualization:** Tom Decroo, Aung Kya Jai Maug, Mohamed Anwar Hossain, Tine Demeulenaere, Nimer Ortuño-Gutiérrez, Bouke C. de Jong, Armand Van Deun.

**Data curation:** Aung Kya Jai Maug, Mohamed Anwar Hossain, Cécile Uwizeye, Mourad Gumusboga, Armand Van Deun.

**Formal analysis:** Tom Decroo, Armand Van Deun.

**Investigation:** Mohamed Anwar Hossain, Cécile Uwizeye, Mourad Gumusboga, Armand Van Deun.

**Methodology:** Tom Decroo, Bouke C. de Jong, Armand Van Deun.

**Project administration:** Aung Kya Jai Maug, Mohamed Anwar Hossain, Armand Van Deun.

**Supervision:** Aung Kya Jai Maug, Mohamed Anwar Hossain, Bouke C. de Jong, Armand Van Deun.

**Validation:** Aung Kya Jai Maug, Mohamed Anwar Hossain, Cécile Uwizeye, Mourad Gumusboga, Tine Demeulenaere, Nimer Ortuño-Gutiérrez, Bouke C. de Jong, Armand Van Deun.

**Writing – original draft:** Tom Decroo, Armand Van Deun.

**Writing – review & editing:** Tom Decroo, Aung Kya Jai Maug, Mohamed Anwar Hossain, Cécile Uwizeye, Mourad Gumusboga, Tine Demeulenaere, Nimer Ortuño-Gutiérrez, Bouke C. de Jong, Armand Van Deun.

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
