## [Decision Letter · Decision Letter 0]

30 Jul 2020

PONE-D-20-18936

Injectables’ key role in rifampicin-resistant tuberculosis treatment outcomes

PLOS ONE

Dear Dr. Decroo,

Thank you for submitting your manuscript to PLOS ONE. Your manuscript was evaluated very positively by an independent referee and myself. Referee #1 raised a couple of minor concerns that should be addressed before we can proceed with formal acceptance. Therefore, we invite you to submit a revised version of the manuscript that addresses the points raised during the review process.

We look forward to receiving your revised manuscript.

Kind regards,

Olivier Neyrolles

Academic Editor

PLOS ONE

2. Thank you for stating in your ethics statement "The evaluation of the STR is covered by a study protocol approved by the National Research Ethics Committee of Bangladesh Medical Research Council and the Institutional Review Board of the Institute of Tropical Medicine Antwerp (1233/18)." Please clarify whether this specific study was fully approved by these ethics committees.

3. Please provide a summary table of patient demographics.

4.We note that you have indicated that data from this study are available upon request. PLOS only allows data to be available upon request if there are legal or ethical restrictions on sharing data publicly. For information on unacceptable data access restrictions, please see http://journals.plos.org/plosone/s/data-availability#loc-unacceptable-data-access-restrictions.

5.Thank you for stating the following in the Financial Disclosure section:

[The authors received no specific funding for this work.]. 

We note that one or more of the authors have an affiliation to the commercial funders of this research study : Independent Consultant

Reviewers' comments:

Reviewer's Responses to Questions

**Comments to the Author**

1. Is the manuscript technically sound, and do the data support the conclusions?

Reviewer #1: Yes

2. Has the statistical analysis been performed appropriately and rigorously? 

Reviewer #1: Yes

3. Have the authors made all data underlying the findings in their manuscript fully available?

Reviewer #1: Yes

4. Is the manuscript presented in an intelligible fashion and written in standard English?

Reviewer #1: Yes

5. Review Comments to the Author

Reviewer #1: The authors demonstrate in scientifically sound and convincing way the important role played by 2nd line injectables in preventing recurrence with acquired resistance to Gatifloxacine, the core drug in the hitherto most effective shorter treatment regimen (STR) of rifampicin resistant tuberculosis (RR-TB). Subsequently, they plead for the necessity to include in the evaluation of all-oral RR-TB regimens - promoted since 2019 by WHO alternatively to STRs including injectables - an assessment of regimen compositions on resistance acquisition. The study is an important contribution to the ongoing discussion about merit and virtue of the standardized ‘Bangladesh’ STR and its variants using injectables for programmatic management of RR-TB in L/MICs where treatment options after recurrence are limited and will be so in a foreseeable future.

Minor suggestions

Title

Consider adding “…in RR-TB STR…”

Discussion

L289-90 Also understandable from the context, consider re-wording (…two days is not

appropriate for all TB drugs.” – “Two days of (what)…”.

6. PLOS authors have the option to publish the peer review history of their article (what does this mean?). If published, this will include your full peer review and any attached files.

Reviewer #1: No

---

## [Author Response · Author response to Decision Letter 0]

31 Jul 2020

Dear Editor,

Many thanks for the review of "Injectables’ key role in rifampicin-resistant tuberculosis shorter treatment regimen outcomes". We have adapted the paper accordingly.

A revised manuscript showing modifications in track changes was uploaded, as well as a clean version and this rebuttal. 

We shall of course be available to respond to any additional question you may have.

Kind regards,

Tom Decroo, on behalf of the co-authors

PONE-D-20-18936

Injectables’ key role in rifampicin-resistant tuberculosis treatment outcomes

PLOS ONE

Dear Dr. Decroo,

Thank you for submitting your manuscript to PLOS ONE. Your manuscript was evaluated very positively by an independent referee and myself. Referee #1 raised a couple of minor concerns that should be addressed before we can proceed with formal acceptance. Therefore, we invite you to submit a revised version of the manuscript that addresses the points raised during the review process.

We look forward to receiving your revised manuscript.

Kind regards,

Olivier Neyrolles

Academic Editor

PLOS ONE

2. Thank you for stating in your ethics statement "The evaluation of the STR is covered by a study protocol approved by the National Research Ethics Committee of Bangladesh Medical Research Council and the Institutional Review Board of the Institute of Tropical Medicine Antwerp (1233/18)." Please clarify whether this specific study was fully approved by these ethics committees.

RESPONSE: The approved protocols cover the objectives, data collection procedures, and analysis of the present study. We adapted the statement as follows: “The evaluation of the STR is fully covered by a study protocol approved by the National Research Ethics Committee of Bangladesh Medical Research Council and the Institutional Review Board of the Institute of Tropical Medicine Antwerp (1233/18).”

3. Please provide a summary table of patient demographics.

RESPONSE: We added a summary table with age and gender, now “table 2” and added a sentence in the results text.

4.We note that you have indicated that data from this study are available upon request. PLOS only allows data to be available upon request if there are legal or ethical restrictions on sharing data publicly. For information on unacceptable data access restrictions, please see http://journals.plos.org/plosone/s/data-availability#loc-unacceptable-data-access-restrictions.

RESPONSE: There are ethical and privacy concerns with sharing the dataset. Given the stigma associated with RR-TB disease and to maximally protect the patient’s privacy we prefer to not share the database in an unrestricted manner. Some specific medical data (combination of a rare outcome + age + gender +++ …) might enable re-identification, even though identifiers such as names and addresses were removed from the programme dataset prior to analysis.

We propose the following statement: “The data supporting the findings of this publication are retained at the Institute of Tropical Medicine, Antwerp and will not be made openly accessible due to ethical and privacy concerns. Data can however be made available after approval of a motivated and written request to the Institute of Tropical Medicine at ITMresearchdataaccess@itg.be”

 5.Thank you for stating the following in the Financial Disclosure section:

[The authors received no specific funding for this work.]. 

RESPONSE: We added this statement under “Funding”.

 We note that one or more of the authors have an affiliation to the commercial funders of this research study : Independent Consultant

RESPONSE: AVD is an expert in the field of TB diagnosis and treatment. He worked for decades for Damien Foundation (DF) in their NGO medical aid projects, in one of which the study was conducted, and the Institute of Tropical Medicine Antwerp (ITM), where samples were send. Neither of those is of a commercial nature. Now he is retired. Given his expertise he still contributes to DF and ITM activities in general. Contribution to this specific study was not specified in the consultancy contract. He has no competing interests and no commercial affiliation. Therefore the statement “The authors received no specific funding for this work” applies to all authors.

Reviewers' comments:

Reviewer's Responses to Questions

Comments to the Author

1. Is the manuscript technically sound, and do the data support the conclusions?

Reviewer #1: Yes

2. Has the statistical analysis been performed appropriately and rigorously? 

Reviewer #1: Yes

3. Have the authors made all data underlying the findings in their manuscript fully available?

Reviewer #1: Yes

4. Is the manuscript presented in an intelligible fashion and written in standard English?

Reviewer #1: Yes

5. Review Comments to the Author

Reviewer #1: The authors demonstrate in scientifically sound and convincing way the important role played by 2nd line injectables in preventing recurrence with acquired resistance to Gatifloxacine, the core drug in the hitherto most effective shorter treatment regimen (STR) of rifampicin resistant tuberculosis (RR-TB). Subsequently, they plead for the necessity to include in the evaluation of all-oral RR-TB regimens - promoted since 2019 by WHO alternatively to STRs including injectables - an assessment of regimen compositions on resistance acquisition. The study is an important contribution to the ongoing discussion about merit and virtue of the standardized ‘Bangladesh’ STR and its variants using injectables for programmatic management of RR-TB in L/MICs where treatment options after recurrence are limited and will be so in a foreseeable future.

RESPONSE: Many thanks for your encouraging feedback

Minor suggestions

Title

Consider adding “…in RR-TB STR…”

RESPONSE: The revised title reads as follows: “Injectables’ key role in rifampicin-resistant tuberculosis shorter treatment regimen outcomes”

Discussion

L289-90 Also understandable from the context, consider re-wording (…two days is not

appropriate for all TB drugs.” – “Two days of (what)…”.

RESPONSE: The text was revised as follows: “Besides EBA estimates being prone to considerable variation, assessment at two days treatment is not appropriate for all TB drugs”

6. PLOS authors have the option to publish the peer review history of their article (what does this mean?). If published, this will include your full peer review and any attached files.

Do you want your identity to be public for this peer review? For information about this choice, including consent withdrawal, please see our Privacy Policy.

Reviewer #1: No

---

## [Editor Report · Decision Letter 1]

10 Aug 2020

Injectables’ key role in rifampicin-resistant tuberculosis shorter treatment regimen outcomes

PONE-D-20-18936R1

Dear Dr. Decroo,

We’re pleased to inform you that your manuscript has been judged scientifically suitable for publication and will be formally accepted for publication once it meets all outstanding technical requirements.

Kind regards,

Olivier Neyrolles

Section Editor

PLOS ONE

---

## [Editor Report · Acceptance letter]

14 Aug 2020

PONE-D-20-18936R1 

Injectables’ key role in rifampicin-resistant tuberculosis shorter treatment regimen outcomes 

Dear Dr. Decroo:

I'm pleased to inform you that your manuscript has been deemed suitable for publication in PLOS ONE. Congratulations! Your manuscript is now with our production department. 

Kind regards, 

on behalf of

Dr. Olivier Neyrolles 

Section Editor

PLOS ONE